

# Profile and clinical implication of circular RNAs in human papillary thyroid carcinoma

Huihui Ren[1,*], Zhelong Liu[1,*], Siyue Liu[1], Xinrong Zhou[1], Hong Wang[2], Jinchao Xu[2], Daowen Wang[1,2] and Gang Yuan[1]

[1] Department of Internal Medicine, Tongji Hospital, Huazhong University of Science and Technology, Wuhan, China
[2] Molecular Diagnostic Laboratory, Tongji Hospital, Huazhong University of Science and Technology, Wuhan, China
[*] These authors contributed equally to this work.

Corresponding authors
Daowen Wang,
dwwang@tjh.tjmu.edu.cn
Gang Yuan,
gangyuan@tjh.tjmu.edu.cn

## ABSTRACT

**Background**. Differently expressed circular RNAs (circRNAs) have been reported to play a considerable role in tumor behavior; however, the expression profile and biological function of circRNAs in papillary thyroid carcinoma (PTC) remains unknown. Thus, the study was aimed to characterize the circRNA expression profile to comprehensively understand the biological behavior of PTC.

**Methods**. We investigated the expression profile of circRNAs using circRNA microarray in three pairs of PTC and adjacent normal tissues. Quantitative real-time reverse transcription polymerase chain reaction (qRT-PCR) was used to validate eight candidate circRNAs in 40 paired PTC tumors and adjacent normal samples. Next, we employed a bioinformatics tool to identify putative miRNA and circRNA-associated downstream genes, followed by constructing a network map of circRNA-miRNA-mRNA interactions and exploring the potential role of the candidate circRNAs.

**Results**. In total, 206 up- and 177 downregulated circRNAs were identified in PTC tissues (fold change >1.5; $P < 0.05$). The expression levels of eight candidate circRNAs confirmed by qRT-PCR were significantly different between the PTC and normal samples. The downstream genes of candidate circRNAs participated in various biological processes and signaling pathways. The most up and downregulated circRNAs were hsa_circRNA_007148 and hsa_circRNA_047771. The lower expression level of hsa_circRNA_047771 was associated BRAFV600 mutation, lymph node metastasis (LNM), as well as with advanced TNM stage (all $P < 0.05$). The higher expression level of hsa_circRNA_007148 was significantly correlated with LNM ($P < 0.05$). The areas under receiver operating curve were 0.876 (95% CI [0.78–0.94]) for hsa_circRNA_047771 and 0.846 (95% CI [0.75–0.96]) for hsa_circRNA_007148.

**Discussion**. The study suggests that dysregulated circRNAs play a critical role in PTC pathogenesis. PTC-related hsa_circRNA_047771 and hsa_circRNA_007148 may serve as potential diagnostic biomarkers and prognostic predictors for PTC patients.

## INTRODUCTION

Thyroid cancer is the most common malignant tumor of the endocrine organs, whose morbidity has continuously increased in recent decades (*La Vecchia et al., 2015*; *Siegel, Miller & Jemal, 2017*). The dominant histological type is papillary thyroid carcinoma (PTC), which accounts for 85–95% of all cases (*La Vecchia et al., 2015*). Although many studies have been performed to better understand PTC, controversy remains regarding the malignancy diagnosis, postoperative treatment, and prognosis in patients with thyroid nodules. Thyroid fine-needle aspiration biopsy (FNAB) is a standard method for the preoperative evaluation of thyroid nodules, up to 30% of nodules with indeterminate cytology; however, it cannot be definitively diagnosed (*Haugen, 2017*). Operative treatment was suggested for most of these patients, although only 10–20% of nodules were diagnosed as malignant lesions. An urgent clinical challenge is to develop an effective diagnostic method to reduce overdiagnosis and overtreatment.

Recent studies have provided evidence that genetic factors may play a key role in the development of thyroid cancer. The BRAFV600E mutation, resulting in activation of MAPK pathway signaling, is significantly associated with more aggressive characteristics of PTCs and is of great importance in risk-stratification and the management of patients with thyroid nodules (*Kwak et al., 2009*; *Xing, 2007*). In addition, various noncoding RNAs (ncRNAs), such as microRNAs (miRNAs) and linear noncoding RNAs (lncRNAs), also participate in the progression and pathogenesis of PTC (*Fuziwara & Kimura, 2016*; *Liyanarachchi et al., 2016*). However, the genomic and epigenomic mechanisms in thyroid cancer pathogenesis have not been clarified.

Circular RNAs (circRNAs), a novel class of endogenous noncoding RNAs, have become a research hotspot in the cancer field. Circular RNAs, unlike linear RNA, have no poly(A) tails and 5′ caps, are characterized by a covalently closed loop structure formed by a tail 3′ splice site and a head 5′ splice site (*Jeck & Sharpless, 2014*). Recently, circRNAs were found to be highly conserved in sequence, maintained stability in mammalian cells, and showed tissue-specific expression (*Chen & Schuman, 2016*; *Rybak-Wolf et al., 2015*). These properties make circRNAs potential molecular biomarkers for multiple diseases. Furthermore, circRNAs are important transcriptional regulators of gene expression involved in the pathogenesis of various diseases. For example, circRNAs could function as miRNA sponges to compete with endogenous RNA, regulating the expression of target genes (*Zhao & Shen, 2015*). A human circRNA, ciRS-7 transcript (antisense to the cerebellar degeneration-related protein), acts as an miRNA-7 sponge by binding to the microRNA effector complexes, which inactivate the function of miRNA (*Lukiw, 2013*). Mounting evidence has demonstrated that circRNAs play a significant role in carcinogenesis, acting as oncogenes or onco-suppressors. Differentially expressed circRNAs were observed in various types of cancers and were associated with the development, invasion and metastasis of human tumors (*Chen et al., 2017*; *Han et al., 2017*; *Meng et al., 2017*; *Wu et al., 2010*). These lines of evidence have suggested that circRNAs can be potential therapeutic targets for diseases.

Although the association between circRNAs and various human cancers has been extensively evaluated and well established, the diagnostic value and biological function of circRNAs in PTC remain largely elusive. Until now, the only study focused on this issue was reported by *Peng et al. (2017)* and included 18 thyroid samples (six PTC, six paired contralateral normal samples, and six benign thyroid lesions). Although a role of circRNAs in PTC pathogenesis was observed, the clinical value of circRNAs for PTC diagnosis, therapy, and prognosis remains undefined. The present study explored the circRNA expression profile in PTC and paired normal tissues and examined the fundamental role of circRNAs in PTC, contributing not only to the understanding of tumorigenesis but also to improving the diagnosis and management of thyroid cancer.

## MATERIALS AND METHODS

### Patients and tissue samples

This study was approved by the Ethics Committee of Tongji Hospital, Tongji Medical, College, Huazhong University of Science and Technology (RB approval number, TJ-C20150806). Written consent was obtained from the patients before surgery. Forty PTC and matched adjacent noncancerous tissue samples were obtained from patients who had undergone thyroidectomy at the Department of General Surgery at Tongji Hospital Affiliated Huazhong University of Science and Technology (Wuhan, Hubei province, China) from September 2016 to February 2017. All tissue samples were immediately soaked in RNAlater Reagent (Servicebio, Wuhan, China) and were preserved at −80 °C until further use. Of these samples, three pairs were used for circRNA microarray analysis, and 40 samples (including the remaining 37 samples and the three samples used for microarray analysis) were used for additional validation in the present study. The general clinicopathologic characteristics of the participants are shown in Table 1. None of the participants included in this study underwent radiotherapy and chemotherapy preoperatively.

### Total RNA isolation and quality control

Total RNA was isolated from PTC and paired adjacent normal tissues using a DNA/RNA coextraction kit (Tiangen Biotech, Beijing, China; DP121221) according to the manufacturer's protocol. The purity and concentration of the total RNA samples were qualified and quantified using the NanoDrop ND1000 system (NanoDrop Technologies/Thermo Scientific, Wilmington, DE, USA). For spectrophotometry, the O.D. A260/A280 ratio should be close to 2.0 for pure RNA. The O.D. A260/A230 ratio should be more than 1.8. The RNA integrity was assessed by electrophoresis using a denaturing agarose gel.

### Microarray analysis

The microarray analysis of circRNAs was performed based on the Arraystar's standard protocols (Arraystar, Rockville, MD, USA). Briefly, total RNA was digested with RNase R (Epicentre, Inc., Madison, WI, USA) to remove linear RNAs and enrich circular RNAs. Next, the enriched circular RNAs were amplified and transcribed into fluorescent cRNA

**Table 1  Clinicopathological characteristics of participates patients.**

| | Patients ($n = 40$) |
|---|---|
| Age (years) | 43.5(13.1) |
| ≥45 | 21(52.5) |
| <45 | 19(47.5) |
| Sex | |
| Female | 30(75) |
| male | 10(25) |
| Hashimoto | |
| Yes | 6(15) |
| No | 34(85) |
| TIRAID | |
| 4a | 4(10) |
| 4b | 13(32.5) |
| 4c | 23(57.5) |
| BRAF mutation | |
| Yes | 16(40) |
| No | 24(60) |
| Tumor size (cm) | 1.2(0.8–2) |
| TNM stage | |
| I | 25(62.59) |
| II | 0 |
| III | 9(22.5) |
| IV | 6(15) |
| LNM | |
| Yes | 18(45) |
| No | 22(55) |
| Focality | |
| Unifocal | 13(32.5) |
| Multifocal | 27(67.5) |

**Notes.**
Data are expressed as mean (sd) or n(percent) or median (interquartile range).
TIRAIDS, thyroid imaging reporting and data system.; TNM, tumor-node-metastasis; LNM, lymph node metastasis.

utilizing a random priming method (Arraystar Super RNA Labeling Kit; Arraystar). The labeled cRNAs were hybridized onto the Arraystar Human circRNA Array V2 (8 × 15K; Arraystar, Rockville, MD, USA). After washing the slides, the arrays were scanned using the Agilent Scanner G2505C system (Agilent, Santa Clara, CA, USA).

## Microarray data analysis

Agilent Feature Extraction software (version 11.0.1.1; Agilent, Santa Clara, CA, USA) was used to analyze acquired array images. Quantile normalization and subsequent data processing were performed using the R software limma package (*Ritchie et al., 2015*). Differentially expressed circRNAs with statistical significance between two groups were identified through volcano plot filtering or fold change filtering. Hierarchical clustering was performed to show the distinguishable circRNA expression patterns among the samples.

The interaction of circRNAs and miRNAs was predicted using Arraystar's homemade miRNA target prediction software based on TargetScan and miRanda. All the differentially expressed circRNAs were annotated in detail using the circRNA/miRNA interaction information.

## Functional analysis

In order to further estimate the biological function of these candidate circRNAs, the targeted genes of miRNAs that were predicted to interact with circRNAs were determined using TargetScan (*Enright et al., 2003*) and miRanda (*Pasquinelli, 2012*). All miRNA gene targets were identified by a *p*-value cutoff at 0.05. These putative target genes were then checked for functional enrichment using DAVID (Database for Annotation, Visualization and Integrated Discovery) (*Huang da, Sherman & Lempicki, 2009*), and KEGG (Kyoto Encyclopedia of Genes and Genomes) analysis was used to identify the significant pathways involved. The circRNA-miRNA-mRNA interaction network was constructed by Cytoscape (http://www.cytoscape.org/) (*Shang et al., 2016*).

## Reverse transcription and qPCR validation

For circRNA analysis, total RNA was reverse transcribed using the FastQuant RT kit (KR106; Tangent Biotech, Beijing, China) and random primers. qRT-PCR was achieved using the Fast SYBR® Green Master Mix (Thermo Fisher Scientific, Waltham, MA, USA) in a StepOne Plus™ Real-Time PCR System (Applied Biosystems, Foster City, CA, USA) with a 20-$\mu$L PCR reaction mixture that included 2 $\mu$L of cDNA, 10 $\mu$L of Fast SYBRGreen Master Mix (2×), 0.2 $\mu$L of forward primer (10 $\mu$M), 0.2 $\mu$L of reverse primer (10 $\mu$M), and 7.6 $\mu$L of nucleus-free water. The reactions were incubated in a MicroAmp™ Fast Optical 96-well reaction plate at 95 °C for 20 s, followed by 40 cycles at 95 °C for 5 s and 60 °C for 34 s. The melting curve with a single peak indicated the specific amplification of the expected fragments. The relative RNA expression level was calculated using the $\Delta$Ct method. Divergent primers rather than the commonly used convergent primers were designed for circular RNA amplification. The sequences of GAPDH and circRNAs primers are listed in Table S1. All experiments were repeated three times.

## BRAFV600E mutation

The BRAFV600E mutation was detected in the tumor tissues of these 40 PTC patients using allelic-specific primer PCR (ASP-PCR). A 201-bp fragment was amplified using a primer sequence designed for BRAF (Table S1). The expected PCR products were separated on agarose gels and visualized by golden view staining, followed by Sanger sequencing validation.

## Statistical analysis

All statistical data were analyzed using Statistical Product and Service Solutions SPSS software 19.0 (SPSS, Chicago, IL, USA) and GraphPad Prism 5.0 (GraphPad Software, La Jolla, CA, USA). The differences in expression of circRNAs between PTC and paired adjacent normal tissues were evaluated using Student's *t* test. The mean circRNA expression level was used as the cutoff value to define the high and low expression of circRNAs. The

correlations between the circRNA levels and clinicopathological factors were analyzed by Student's $t$-test, Fisher's exact test, or nonparametric test. A receiver operating curve (ROC) was established to assess the diagnostic performance of hsa_circRNA_047771 and hsa_circRNA_007148. A $P$ value <0.05 was considered significant.

## RESULTS

### circRNA profiles in normal and tumor tissues

In order to profile circRNA expression in PTC, we performed circRNA microarray analysis in PTC and matched noncancerous thyroid tissues. Hierarchical clustering analysis indicated that the expression patterns of circRNAs in PTC tissues were significantly different from those in normal tissues (Fig. 1A, fold change >1.5, $p < 0.05$). Red and green colors indicate the high and low expression levels of circRNAs, respectively. The red point of the volcano plot determined the significantly differentially expressed circRNAs between the normal and PTC samples with a fold change >1.5 (Fig. 1B). The scatter plot revealed the difference in circRNA expression in PTC tumor tissues and paired normal tissues (Fig. 1C). The points above and below the green lines represent 1.5-fold up and down of circRNAs between the normal and PTC samples, respectively. We found that 383 circRNAs were differentially expressed in PTC tissues compared with normal tissues. Among them, 206 circRNAs were up regulated and 177 circRNAs were downregulated (fold change ≥1.5, $P < 0.05$ and FDR <0.05) in three paired carcinoma and normal tissues. The differently deregulated circRNAs were mostly located in exons (Fig. 1D). The top 10 differentially expressed circRNAs are listed in Table 2.

### Validation of dysregulated circRNAs

According to the fold change tested in the microarray analysis, eight circRNAs (four upregulated and four downregulated) were selected for validation of the microarray results in 40 paired nontumorous and tumor samples by qRT-PCR. All four upregulated and four downregulated circRNAs were differentially expressed in PTC compared with those in normal tissues (Figs. 2A–2H). These results were consistent with the microarray data, indicating the validation of these results.

### Functional analysis of differentially expressed circRNAs

It is widely accepted that circRNAs are important transcriptional regulators of gene expression involved in the pathogenesis of various cancers. To explore the underlying mechanisms of dysregulated circRNAs involved in the tumorigenesis or development of PTC, Gene Ontology (GO) enrichment analysis of differentially expressed target genes was performed to evaluate the functional significance of these candidate circRNAs. The predominantly enriched GO items of upregulated circRNAs in PTC were correlated with cell communication-related genes (Fig. 3A), while the GO terms of downregulated circRNAs were mainly involved in the plasma membrane (Fig. 3B). These enriched terms, involving cell communication, signal transducer activity, RNA polymerase II transcription factor, and transcriptional activator activity, suggest that these circRNAs play important roles in cell-to-cell signaling and interaction, cellular proliferation, cellular function and maintenance.

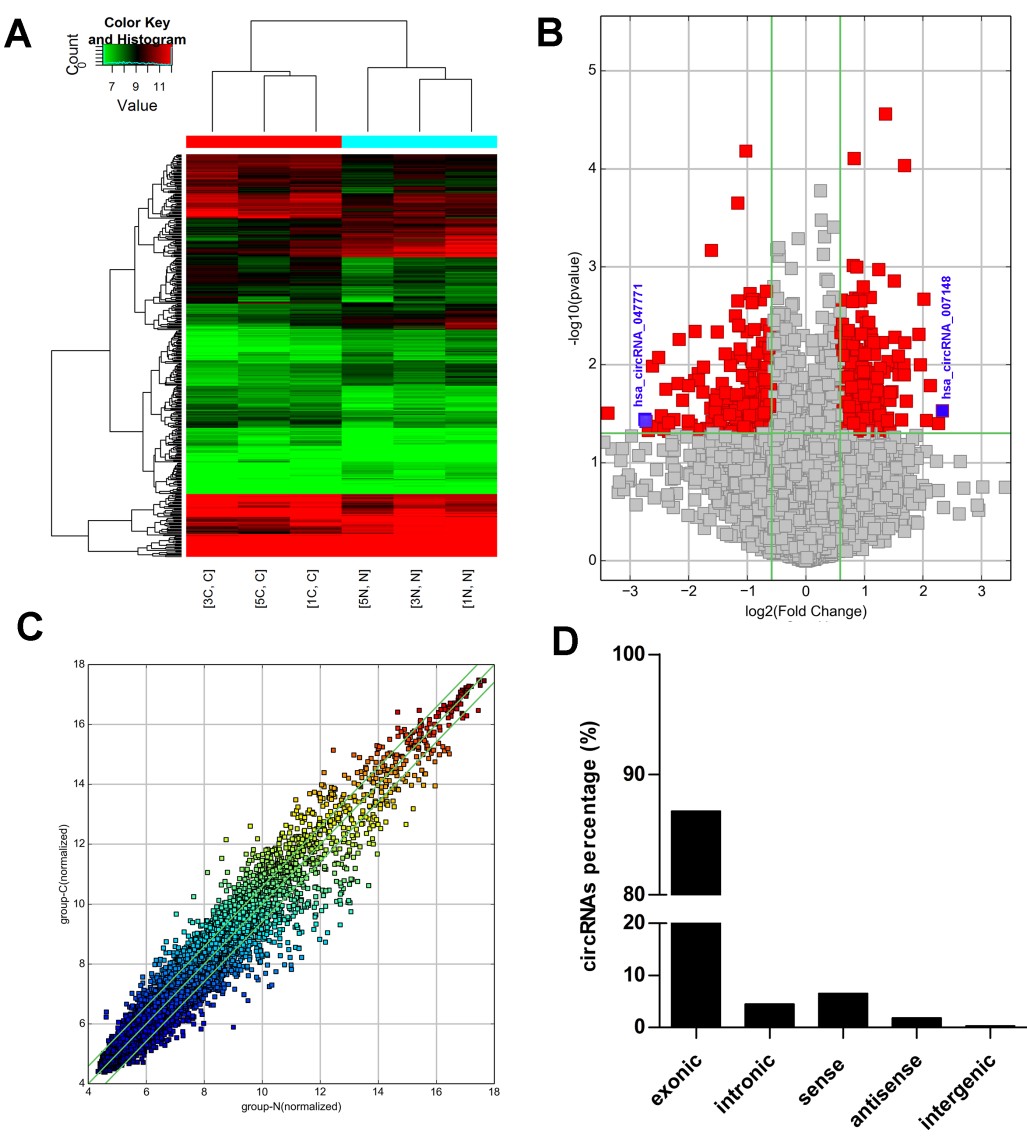

**Figure 1** **circRNAs expression profile in papillary thyroid carcinoma and paired normal tissues.** (A) Heat map from the microarray analysis of circRNA expression in normal and p apillary thyroid carcinoma (PTC) tissues (fold change > 1.5, $p < 0.05$). Red and green colors indicate high and low expression levels, respectively. (B) Scatter plots represent the differential expression of circRNAs in PTC tumors (group-C) versus normal thyroid tissue (group-N). The values of the $x$- and $y$-axes in the scatter plot were the normalized signal intensities of the samples (log2 scaled). The green lines represent the 1.5-fold up and down of circRNAs between the normal and PTC samples. C, PTC tissue. N, adjacent normal tissue. (C) Volcano plot constructed to visualize the differentially expressed circRNAs between PTC and adjacent normal tissues. The vertical lines indicate 1.5-fold upregulation and downregulation. The horizontal line represents a $P$-value of 0.05. The red points in the plot represent the statistically deregulated circRNAs. (D) the proportion of differentially expressed circRNAs according to the different genomic locus (exonic, intronic, antisense, intragenic, and intergenic).

**Table 2  Top 10 most upregulated and downregulated circRNAs in PTC tumors versus paired normal thyroid tissue.**

| circRNA | FC | *P* value | Type | chrom | GeneSymbol | miRNA binding sites |
|---|---|---|---|---|---|---|
| upregulated | | | | | | |
| hsa_circRNA_007148 | 5.014 | 0.02933 | exonic | chr3 | FNDC3B | hsa-miR-6785-5p,hsa-miR-1275,hsa-miR-490-3p,hsa-miR-1229-5p,hsa-miR-6088 |
| hsa_circRNA_004662 | 4.80 | 0.03966 | exonic | chr6 | SOD2 | hsa-miR-4520-2-3p,hsa-miR-208b-5p,hsa-miR-4713-5p,hsa-miR-4659b-3p |
| hsa_circRNA_046843 | 4.36 | 0.01622 | exonic | chr18 | ANKRD12 | hsa-miR-4524a-5p,hsa-miR-4753-3p,hsa-miR-3911,hsa-miR-876-5p,hsa-miR-942-5p |
| hsa_circRNA_103514 | 4.16 | 0.03694 | exonic | chr3 | FNDC3B | hsa-miR-147b,hsa-miR-510-5p,hsa-miR-15b-3p,hsa-miR-1323,hsa-miR-552-3p |
| hsa_circRNA_061346 | 4.03 | 0.00213 | exonic | chr21 | APP | hsa-miR-4778-3p,hsa-miR-5196-3p,hsa-miR-5193,hsa-miR-877-3p,hsa-miR-103a-2-5p |
| downregulated | | | | | | |
| hsa_circRNA_404959 | 10.40 | 0.03101 | exonic | chr12 | CLSTN3 | hsa-miR-6848-5p,hsa-miR-6081,hsa-miR-6720-5p,hsa-miR-5591-5p,hsa-miR-6165 |
| hsa_circRNA_047771 | 6.71 | 0.03596 | exonic | chr18 | NARS | hsa-miR-522-3p,hsa-miR-224-3p,hsa-miR-4677-5p,hsa-miR-140-5p,hsa-miR-153-5p |
| hsa_circRNA_405498 | 6.63 | 0.03760 | exonic | chr16 | GCSH | hsa-miR-6754-5p,hsa-miR-6812-5p,hsa-miR-5572,hsa-miR-6855-5p,hsa-miR-3612 |
| hsa_circRNA_101408 | 6.45 | 0.04666 | exonic | chr14 | TTLL5 | hsa-miR-136-5p,hsa-miR-500a-5p,hsa-miR-578,hsa-miR-34b-5p,hsa-miR-449c-5p |
| hsa_circRNA_014213 | 6.15 | 0.01028 | exonic | chr1 | SPRR1B | hsa-miR-6812-5p,hsa-miR-6819-5p,hsa-miR-608,hsa-miR-5572,hsa-miR-5088-5p |

**Notes.**

circRNA, circular RNA; PTC, papillary thyroid carcinoma; FC, fold change; Chrom, chromosome; miRNA, microRNA.

KEGG analysis revealed that 50 pathways were significantly enriched among the target genes of upregulated circRNAs (Bonferroni <0.05), whereas 50 pathways were associated with downregulated circRNAs (Bonferroni <0.05). Notably, several important pathways were enriched, such as the VEGF signaling pathway, ras signaling pathway, and Notch signaling pathway (Figs. 4A, 4B). These findings suggest that circRNAs might play a significant role in PTC carcinogenesis and metastasis.

## Clinical implications of novel circRNAs in PTC

Based on the validation data, hsa_circRNA_007148 and hsa_circRNA_047771 were the most significantly elevated and decreased in PTC samples compared with those in normal samples. Therefore, we evaluated their potential clinical value by analyzing the correlation of these two candidate circRNAs with several clinicopathological parameters of PTC patients. Table 3 shows that lower hsa_circRNA_047771 expression levels were associated with BRAFV600 mutation, lymph node metastasis (LNM), and TNM stage (all $P < 0.05$). However, we observed no association between the hsa_circRNA_047771 expression levels and other clinicopathological factors, including sex, age, a history of Hashimoto's thyroiditis, thyroid imaging reporting and data system (TIRAIDS), tumor size, and multifocality. We further found that a higher expression level of hsa_circRNA_007148 was significantly correlated with LNM ($P < 0.05$), while no other clinicopathological factor was found to be significantly associated with the hsa_circRNA_007148 expression level

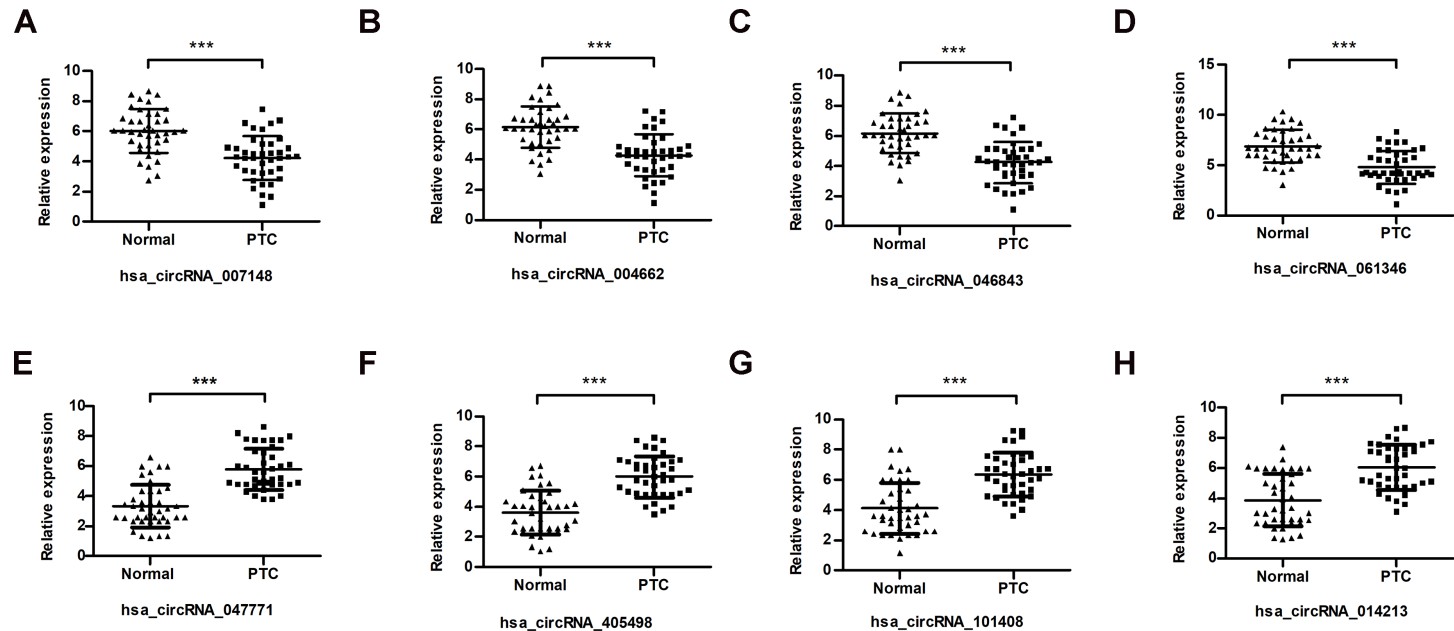

**Figure 2** **Identification and qRT-PCR validation of differentially expressed circRNAs.** (A–D), (E–H) The relative expression of four upregulated and four downregulated circRNAs was validated in 40 human PTC tissues and adjacent non-tumorous tissues. ***$P < 0.001$. Normal: adjacent non-tumorous tissue; PTC: papillary thyroid tumor tissue. ΔCt values were used to assess the relative gene expression, which was normalized to the GAPDH expression level.

($P > 0.05$). Additionally, similar associations were found with the other six deregulated circRNAs (Table S2). Next, we performed ROC curve analysis to assess the diagnostic value of hsa_circRNA_047771 and hsa_circRNA_007148 in distinguishing PTC tissues from normal tissues. The areas under ROC curve were 0.876 (95% CI [0.78–0.94]) for hsa_circRNA_047771 and 0.846 (95% CI [0.75–0.96]) for hsa_circRNA_007148 (Figs. 5A, 5B; Table 4).

## Bioinformatics analysis of hsa_circRNA_047771 and hsa_circRNA_007148

We next used Arraystar's circRNA target prediction software to identify potential miRNAs that could bind to hsa_circRNA_047771 and hsa_circRNA_007148 via miRNA response elements (MREs). The top five ranking MRE targets for the differentially expressed circRNAs are shown in Figs. S1, S2. Notably, among the list of miRNAs targeted by hsa_circRNA_047771, miR-522-3p/miR-153-5p were implicated in PTC in previous reports. We further detected the expression levels of miR-522-3p/miR-153-5p in the 40 paired PTC and normal samples. miR-522-3p/miR-153-5p were upregulated in PTC tissues compared with those in adjacent normal tissues (Fig. S3). These results provide powerful evidence that hsa_circRNA_047771/ miR-522-3p/miR-153-5p could participate in the pathogenesis of PTC. In order to fully understand the underlying mechanism between deregulated circRNAs and the biological progress of PTC, the target prediction program was performed to predict the potential target genes in the database. Based on the data of the

A

B

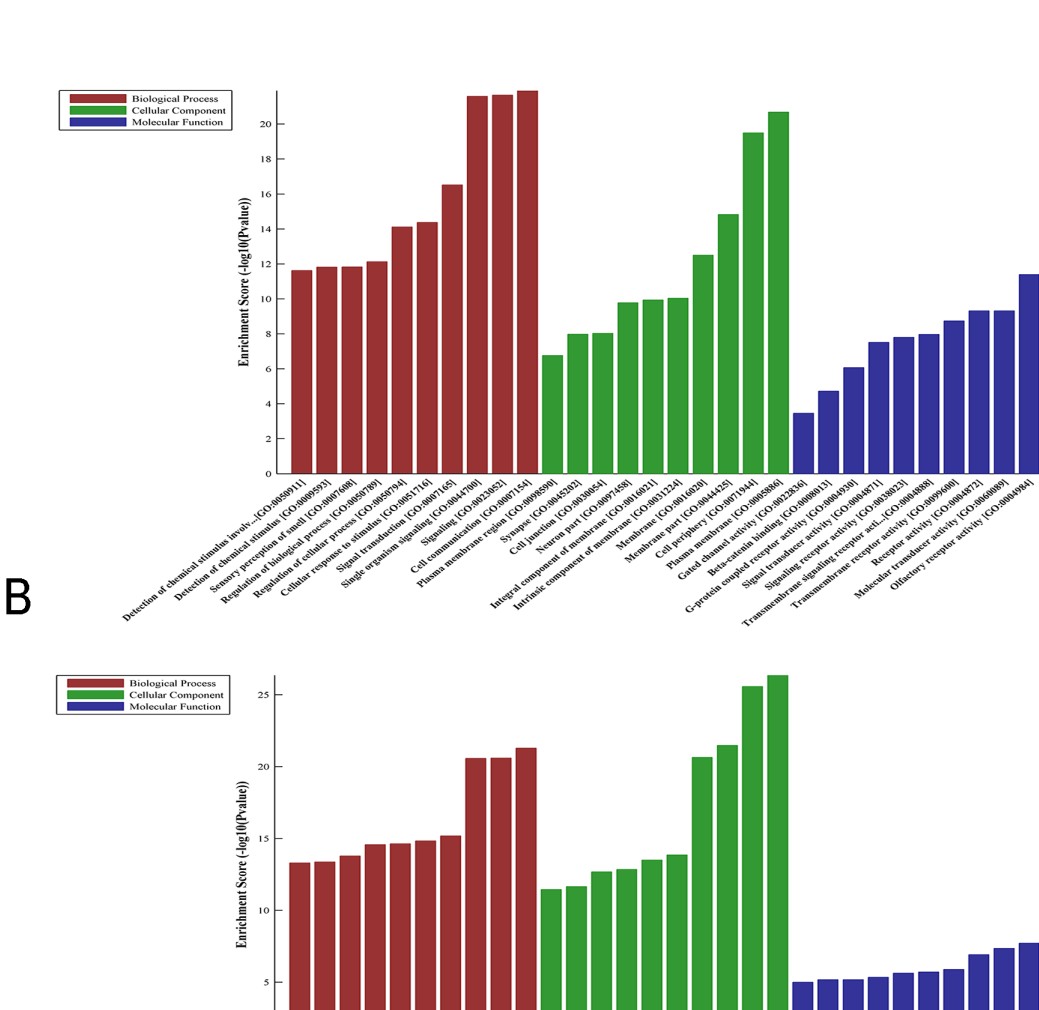

**Figure 3** **Enrichment analysis of circRNA-target genes.** Top 10 significant enriched GO terms of upregulated circRNAs (A) and downregulated circRNAs (B).

deregulated circRNAs, their predicted MREs and targeted genes, we constructed a network map of cancer related circRNA-miRNA-mRNA interactions using Cytoscape that included 36 mRNAs and 34 miRNAs for hsa_circRNA_047771 and 41 mRNAs and 51 miRNAs for hsa_circRNA_007148 (Figs. 6A, 6B).
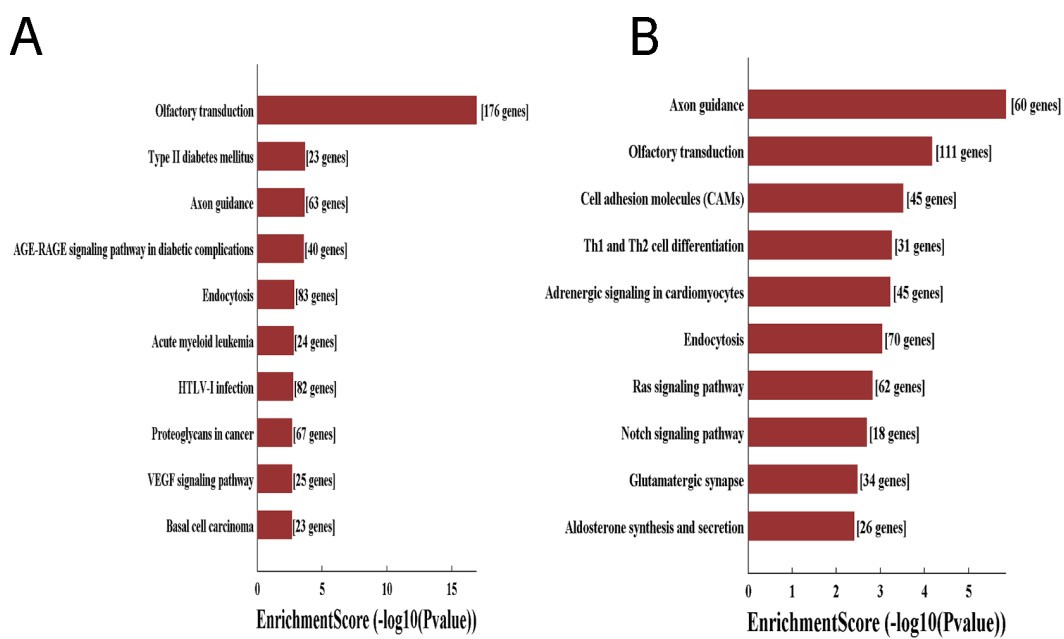

**Figure 4** **Enrichment KEGG analysis of circRNA-target genes.** Top 10 pathway terms in KEGG pathway analysis of upregulated circRNAs (A) and downregulated circRNAs (B).

## DISCUSSION

Recently, the crucial role of epigenetic regulation in the development and pathogenesis of cancers has been widely recognized. There is abundant evidence suggesting that ncRNAs, including miRNAs, lncRNAs and circular RNAs, are strongly associated with the biological behavior of tumors. However, the role of circRNAs in PTC remains largely unknown. In the present study, we characterized the expression profile of circRNAs in PTC tissues and identified many aberrantly expressed circRNAs in PTC tissues, followed by validation of eight significantly different expressed circRNAs. Bioinformatics analysis indicated that deregulated circRNAs might be implicated in the biological process of PTC by regulating tumor-related pathways. The association between two novel circRNAs and aggressive clinicopathological characteristics provide evidence that circRNAs may serve as prognostic predictors for PTC patients. Additionally, the areas under ROC curve suggest that hsa_circRNA_047771 and hsa_circRNA_007148 may be candidate diagnostic biomarkers for PTC.

PTC accounts for a large proportion of thyroid cancer and needs to be better characterized and understood. Although the functions of circRNAs in carcinogenesis and cancer development have drawn great attention, studies focused on the role of circRNAs in PTC remain scarce. To our knowledge, only one study has focused on the relationship between circRNAs and PTC (*Peng et al., 2017*). This study performed circRNA microarray on six paired PTC tumors and contralateral normal tissues, as well as six benign thyroid nodules, to investigate the profile of circRNA expression in PTC and identified one significantly downregulated circRNA (hsa_circRNA_100395) that was

**Table 3 Association between hsa_circRNA_047771 and hsa_circRNA_007148 expression levels and clinicopathological characteristics in 40 PTC patients.**

| | hsa_circRNA_047771[a] | | hsa_circRNA_007148[a] | |
|---|---|---|---|---|
| | High expression | Low expression | High expression | Low expression |
| Age (years) | | | | |
| ≥45 | 8(42.1) | 13(61.9) | 11(64.7) | 10(43.5) |
| <45 | 11(57.9) | 8(38.1) | 6(35.3) | 13(56.5) |
| Sex | | | | |
| Female | 14(73.7) | 16(76.2) | 13(76.5) | 17(73.9) |
| male | 5(26.3) | 5(23.8) | 4(23.5) | 6(26.1) |
| Hashimoto | | | | |
| Yes | 3(15.8) | 3(14.3) | 2(10.5) | 4(19.0) |
| No | 16(84.2) | 18(85.7) | 15(88.2) | 19(82.6) |
| TIRAIDS | | | | |
| 4a | 2(10.5) | 2(9.5) | 2(11.8) | 2(8.7) |
| 4b | 7(36.5) | 6(28.6) | 4(23.5) | 9(39.1) |
| 4c | 10(52.6) | 13(61.9) | 11(64.7) | 12(52.2) |
| BRAF mutation | | | | |
| Yes | 5(25) | 12(60)* | 8(47.1) | 8(34.8) |
| No | 15(75) | 8(40) | 9(52.9) | 15(65.2) |
| Tumor size (cm) | 1.2(0.8–2.0) | 1.3(1–2.05) | 1.7(1.4–2.3) | 1.3(0.8–2.0) |
| TNM stage | | | | |
| I–II | 16(84.2) | 9(42.9)** | 7(41.2) | 18(78.3)* |
| III–V | 3(15.8) | 12(57.1) | 5(58.8) | 10(21.7) |
| LNM | | | | |
| Yes | 7(36.8) | 15(71.4)* | 11(64.7) | 11(48.7) |
| No | 12(63.2) | 6(28.6) | 6(35.5) | 12(52.2) |
| Focality | | | | |
| Unifocal | 14(73.7) | 18(85.7) | 11(64.7) | 16(69.9) |
| Multifocal | 5(26.3) | 3(14.3) | 6(35.5) | 7(30.4) |

**Notes.**

[a] The mean expression value was used as the cutoff scores for the high-expression and low-expression of circRNAs. Data are expressed as n (percent) or median (interquartile range).

*$P < 0.05$.

**$P < 0.01$.

circRNA, circular RNA; PTC, papillary thyroid carcinoma; TIRAIDS, thyroid imaging reporting and data system; TNM, tumor-node-metastasis; LNM, lymph node metastasis.

associated with two tumor-associated miRNA clusters (miR-141-3p and miR-200a-3p), revealing a potential role of circRNAs in PTC pathogenesis. However, to date, no study has investigated the clinical value of circRNAs in PTC patients. The association between circRNA expression and prognosis factors of PTC, such as tumor size, tumor stage, LNM, and BRAF mutation, remains unclear. It is crucial to understand the underlying mechanism of circRNAs regarding tumor behavior at the molecular level with a larger sample. It will be of great importance to identify novel diagnostic, prognostic, and therapeutic targets. Thus, we performed circular RNA array to characterize the circRNA expression profile to comprehensively understand its clinical implications in PTC. Furthermore, similar to
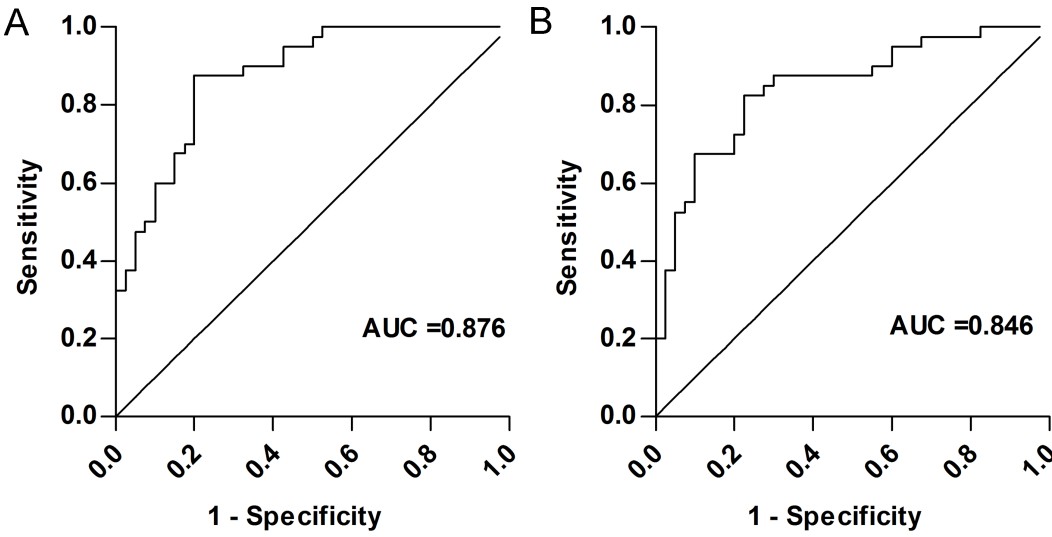

**Figure 5** **Correlation of hsa_circRNA_047771 and hsa_circRNA_007148 with the diagnosisof papillary thyroid carcinoma patients.** Receiver operating characteristics (ROC) curve analysis for hsa_circRNA_047771(A) and hsa_circRNA_007148) (B) according to their relative expression level as measured by qPCR ($P < 0.01$).

**Table 4** **Diagnostic value of circRNAs in papillary thyroid cancer.**

|  | hsa_circRNA_047771 | hsa_circRNA_007148 |
|---|---|---|
| Sensitivity(95% CI) | 87.5(0.73–0.96) | 82.5(0.63–0.97) |
| Specificity(95% CI) | 80.0(0.64–0.91) | 77.5(0.62–0.89) |
| +PV (95% CI) | 81.4(0.66–0.92) | 78.6(0.63–0.90) |
| −PV(95% CI) | 86.5(0.71–0.96) | 81.6(0.66–0.92) |
| Youden index | 0.675 | 0.6 |
| AUC(95% CI) | 0.876(0.78–0.94) | 0.846(0.75–0.96) |

**Notes.**
The relative expression of each circRNA was calculated using the ∆Ct method.
PV, predictive value; AUC, area under the curve; CI, confidence interval.

Nianchun (*Peng et al., 2017*), our study identified deregulated circRNAs. In the present study, KEGG and GO enrichment analyses identified many significant terms involved in cell events and several important cancer-related signaling pathways, such as the VEGF signaling pathway, ras signaling pathway, and Notch signaling pathway. Particularly, the ras signaling pathway was implicated in the development of PTC. These findings suggest that circRNAs might play a significant role in PTC carcinogenesis and metastasis.

Our study identified two novel circRNAs that were significantly associated with aggressive prognostic features of PTC, such as LNM and advanced TNM stage. These data indicated that deregulated circRNAs could predict the prognosis of patients with PTC. Interestingly, we found that a lower expression of hsa_circRNA_047771 is more likely to manifest with BRAFV600E mutation. BRAFV600E mutation has emerged as a poor prognostic marker for PTC patients. The clinical usefulness of BRAFV600E in the risk

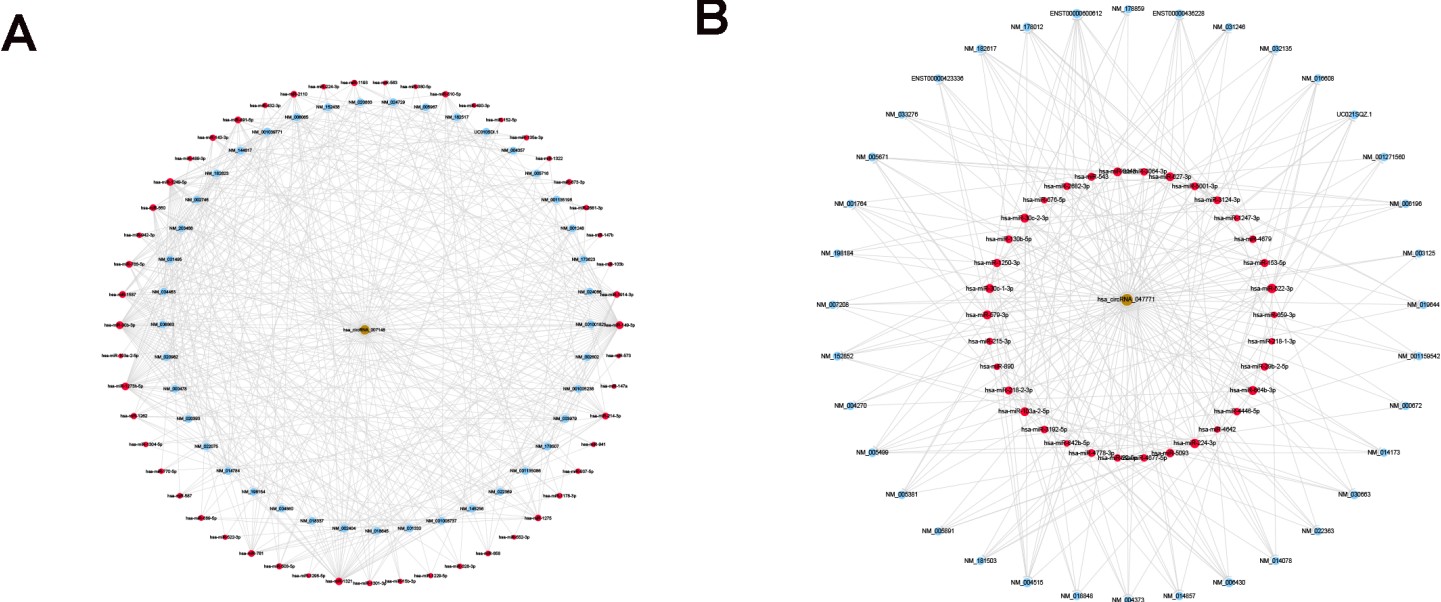

**Figure 6    Construction of the circRNA-miRNA-mRNAnetwork.** (A) The network map contained the significantly upregulated hsa_circRNA_007148 (represented by brown nodes) along with their 51 downstream miRNAs (represented by red nodes) and 41 target mRNAs (represented by light-blue nodes). (B) The network map contained significantly downregulated hsa_circRNA_47771 (represented by brown nodes) along with their 34 downstream miRNAs (represented by red nodes) and target 36 mRNAs (represented by light-blue nodes).

stratification, diagnosis and management of PTC patients has been established (*Elisei et al., 2008*; *Lupi et al., 2007*). The predictive effect of circRNAs in the prognosis of PTC patients and its association with BRAFV600E mutation in PTC patients are considerably important and need further investigation.

The clinical application of circRNAs as potential molecular markers for the diagnosis and treatment of disease are promising (*Meng et al., 2017*), as confirmed by our results. circRNAs are highly stable and conservative in the cytoplasm and other body fluids, such as saliva and exosomes (*Bahn et al., 2015*; *Dou et al., 2016*). The observed high stability of circRNAs suggests that circRNAs will likely be a reliable biomarker for the preoperative diagnosis of PTC. Given the diagnostic limitations of FNAB in indeterminate cytology and insufficient sensitivity of molecular testing (*Haugen et al., 2016*; *Li et al., 2015*), the detection of circRNAs and clinical significance of circRNAs used for the preoperative diagnosis of thyroid cancers are attractive and critical issues related to thyroid cancer study that need to be well defined.

Circular RNAs are endogenous ncRNA molecules that function as regulators involved in various biological courses of cancer, such as development and proliferation (*He et al., 2017*; *Xin et al., 2017*). The mechanisms underlying circRNAs and tumorigenesis may be primarily associated with altered gene expression mediated by acting as an miRNA sponge (*Chen, Chen & Chuang, 2015*). Recent studies have revealed that aberrant circRNA expression was closely correlated with various cancers, including hepatocellular carcinoma, colon cancer, breast cancer, esophageal cancer, and oral cancer (*Chen et al., 2017*; *Han et*

*al., 2017*; *Hsiao et al., 2017*; *Lu et al., 2017*). *Chen et al. (2017)* found that circRNA_100290 could regulate oral squamous cell carcinomas through miR-29b-mediated inhibition of cell proliferation and the reduction of CDK6 expression. Another circRNA identified in a recent study, termed circHIPK3, was observed to be differentially expressed in several normal and tumor tissues and functions as a gene regulator by regulating miRNA activity (*Zheng et al., 2016*). These findings were confirmed in our study. We observed that each circRNA could bind to multiple miRNAs, some of which have been reported to be involved in PTC biology. For example, we found that miR-522-3p/miR-153-5p were targets of one novel down-expressed circRNA (hsa_circRNA_047771). These deregulated miRNAs have been reported to significantly correlate with thyroid cancer differentiation and progression (*Boufraqech, Klubo-Gwiezdzinska & Kebebew, 2016*; *Fuziwara & Kimura, 2016*). Thus, further investigations are necessary to clarify the specific mechanism between deregulated circRNAs and PTC pathology.

Several limitations of this study must be addressed. First, the sample size was limited, and other thyroid carcinomas were not included in this study other than PTC. Second, we did not clarify the mechanism of circRNA/miRNA in the PTC pathological process. The molecular mechanism should be experimentally determined and elaborated in the future. Third, given the clinical applications of circRNAs, further studies will be required to evaluate the diagnostic value of circRNA levels in serum or FNAB samples.

## CONCLUSIONS

The study demonstrated that several circRNAs are expressed differently in PTC and participate in tumor pathogenesis. Two PTC-associated circRNAs could serve as potential diagnostic markers and predict the prognosis of PTC patients. Furthermore, the study revealed that circRNAs may have great potential as prognosis predictors and therapeutic targets for PTC patients in the near future.

### Funding

This work was supported by the grant from the National Natural Science Foundation of China (No. 81370941, Gang Yuan), Graduates' Innovation Fund, Huazhong University of Science and Technology (No. 5003540003, Huihui Ren) and Thyroid Research Program of Young and Middle-aged Physicians (No.2016-N-07, Zhelong Liu). The funders had no role in study design, data collection and analysis, decision to publish, or preparation of the manuscript.

### Grant Disclosures

The following grant information was disclosed by the authors:
National Natural Science Foundation of China: 81370941.
Graduates' Innovation Fund, Huazhong University of Science and Technology: 5003540003.
Thyroid Research Program of Young and Middle-aged Physicians: 2016-N-07.

## Competing Interests

The authors declare there are no competing interests.

## Author Contributions

- Huihui Ren, Zhelong Liu, performed the experiments, analyzed the data, prepared figures and/or tables, authored or reviewed drafts of the paper, approved the final draft.
- Siyue Liu and Xinrong Zhou performed the experiments, analyzed the data, contributed reagents/materials/analysis tools, prepared figures and/or tables, authored or reviewed drafts of the paper, approved the final draft.
- Hong Wang and Jinchao Xu performed the experiments, contributed reagents/materials/analysis tools, prepared figures and/or tables, authored or reviewed drafts of the paper, approved the final draft.
- Daowen Wang and Gang Yuan conceived and designed the experiments, authored or reviewed drafts of the paper, approved the final draft.

## Human Ethics

The following information was supplied relating to ethical approvals (i.e., approving body and any reference numbers):

The Ethics Committee of Tongji Hospital, Tongji Medical College, Huazhong University of Science and Technology granted approval to carry out the study within its facilities (RB approval number, TJ-C20150806).

## Microarray Data Deposition

The following information was supplied regarding the deposition of microarray data:

The microarray is available as a Supplemental File.

## Data Availability

The raw data are provided in Supplemental Files.

## Supplemental Information

Supplemental information for this article can be found online at http://dx.doi.org/10.7717/peerj.5363#supplemental-information.

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
