# Peer review of "Profile and clinical implication of circular RNAs in human papillary thyroid carcinoma"

_PeerJ, doi:10.7717/peerj.5363_

## Round 0.1 · original submission · Major Revisions

Dear Dr Gang Yuan

Please address all the reviewers comments and suggested modifications.
In addition please proceed to the following changes:

The sentence “Recent researches have provided evidence that thyroid cancer is widely considered to be a genetic disease.” Is inaccurate (in fact all cancers are long known to be genetic diseases in this sense) Please delete or rephrase.

Please clarify the number of samples used in the different experiments ” Of these samples, three pairs were used for circRNA microarray analysis and 40 samples (including samples used for microarray analysis) were used for additional validation in the present study.” Have the authors used 43 samples or 3+37=40?

Please correct “All results were repeated for three times.” by
“All experiments were repeated three times.” If it is the case, of course.

In Line 168 please specify the number of cases and normal thyroids used.

A more complete description of the results is necessary. The results are superficially mentioned, always referring to the paper Figures, The authors must do a description of the results in the Results section.

Figure 3 a) and b) must be remade since it is not possible to read the small and fuzzy letters .

The MS needs a through and professional revision of the English. There are numerous errors and misleading phrases.

Reviewer 1 ·

Basic reporting

The manuscript lacks clarity and professional writing. The English in the manuscript should be corrected before publication.

Experimental design

The investigation is otherwise meaningful and OK, but I would like to point out one case in which the authors should provide more details. In Results section, under the subsection "Functional analysis of differentially expressed circular RNAs", the identification of the target genes of the circular RNAs, which has been used for GO and pathway enrichment analysis, is not clear. Are they identified as the target genes of the miRNAs that are predicted o target the circular RNAs? The authors never mentioned anything about this. And how did they identify differentially expressed genes? The methods section only describes microarray experiments for circular RNAs with Arraystar Human circRNA Array V2, which is not for standard protein coding genes. So it totally unclear how the authors detected differentially expressed genes. I request the authors to explain this.

Validity of the findings

In the functional analysis section (page 15-16), the authors reported the results of the enrichment analysis for GO and KEGG pathways for the upregulated or downregulated circular RNAs. However, as I mentioned already, how they got the target gene set for these analysis is not clear. Moreover, the authors did not elaborate further what are the implications of the enriched processes or pathways in context to PTC. The authors should either provide sufficient explanation or remove this section.

Additional comments

The study reports two circular RNAs to be associated with clinical factors in PTC. They chose these two circular RNAs in the basis that these were the top upregulated or downregulated candidate between PTC/normal tissues. They did not mention if there were other candidate circular RNAs in their study which had correlation with clinical factors. Did the authors check other circular RNAs from their list for potential clinical relevance or they just picked these two circular RNAs? It is not mentioned in the Results or Discussion section. If they did not include other disregulated circular RNAs then why so? I would recommend they search for the clinical implications of other circular RNAs as well, if not done already.

Reviewer 2 ·

Basic reporting

The writing is not of professional standards. I suggest the authors check the English by a competent person in the next version

Experimental design

The study aims to find out circular RNAs associated with Papillary Thyroid Carcinoma (PTC). They first did microarray in three pairs of patient samples from PTC and adjacent normal tissues to identify differentially expressed circular RNAs and then they validated eight of them with RT-PCR in 40 paired PTC/normal samples. Finally, they associated expression of two circular RNAs with clinico-pathological factors. Additionally, they showed the possible miRNA interactions of these two circular RNAs with bioinformatic analysis. I have two things to comment on the experimental design process. Firstly, was it only the two circular RNAs they reported that had any association with clinical factors? If that was the case it would be fine, but the authors never mentioned if they tried to associate other circular RNAs with clinical factors and did not find significant association. Secondly, the authors reported that by bioinformatic analysis they found miR-522-3p/miR-153-5p, which have been implicated in PTC, to have putative target site in one of the clinically relevant circular RNA hsa_circRNA_047771. It would be nice if they could validate the expressions of these miRNAs in their samples and correlate the expressions of hsa_circRNA_047771 and miR-522-3p/miR-153-5p to further develop on this point.

Validity of the findings

The section on functional analysis of circular RNAs with KEGG and GO enrichment analysis looks redundant and does not provide any solid information in context with the potential association of the circular RNAs with PTC. The authors should clearly explain what is the relevance of this study.

Additional comments

none

---

## Round 0.2 · Major Revisions

Dear Dr Gang,

Although the paper was improved there remains some important points to be corrected, suggested by the reviewers. Please fully address all the remarks

Reviewer 1 ·

Basic reporting

The English writing in the manuscript has been sufficiently improved.

Experimental design

In this revised version, the descriptions in the Materials and Methods section has been provided with sufficient clarity. However, there is still a misconception which should be cleared. In the subsection Functional analysis, the authors described that they performed GO and pathway enrichment analysis on the "differentially expressed" target genes of circRNAs. This is a mis-representation. Because from the study it is evident that the authors did not check the expression of the target genes, they only predicted the target genes using in-silico miRNA-target prediction tools. So, in line 133, instead of writing "These differentially expressed genes were then enriched using DAVID", the correct way to represent is "These putative target genes were then checked for functional enrichment using DAVID". This representation should be corrected in the subsequent sections also.

Validity of the findings

The authors have included some explanations on the relevance of the functional enrichment analysis of the circular RNAs in context to cancer in general; though not conclusive in context of PTC. But there remains another major point which the authors did not address in the revised version: they only checked the association of two circular RNAs with the clinical parameters. While I understand that these two circular RNAs had the most fold-change in expression, still I feel that this study will be incomplete if the authors do not check the association of other up-regulated/down-regulated circRNAs with clinical parameters. At least for the eight circular RNAs whose expression changes were validated by qRT-PCR in 40 paired tumor/normal samples, checking the association with clinical parameters should not be too much work.

Reviewer 3 ·

Basic reporting

This study by Ren H. et al. aimed to profile the circRNA expression and evaluate its functional consequence and clinical implication in papillary thyroid carcinoma. The manuscript is well written and the hypothesis has a strong basis as growing evidence show the functional involvement of ncRNA’s including circRNA
in cancers.

Experimental design

Overall the experiments designed has met with the conclusion, but may require below modification/evidence.

1. The scatter plot (Fig. 1B) would convey better if you indicate/label the candidate circRNA (hsa_circRNA_047771 and _007148 ) within the plot itself.
2. Correct the spelling for “exonic” in Fig. 1D
3. The output from the GO analysis is very general and so it did not add much to the conclusion, however the KEGG pathway analysis does show some relevant pathways (RAS, NOTCH, VEGF) associated with tumorigenesis suggests it could be improved. To further strengthen and to obtain more specific functional pathways, if feasible with your data, I would recommend doing Gene Set Enrichment Analysis (GSEA) or similar type analysis.
4. In Table 3, column hsa_circRNA_047771 under the Row BRAF mutation, the percentage within the bracket (both low and high expression) doesn’t tally up to 100 %? confirm/correct.
5. Figure 6 is overloaded, if it could be simplified/add a subfigure, with only the significant/major cancer associated MRE network, it will better convey the message.
6. The hsa_circRNA_047771/miR-522-3p/miR-153-5p role was not sufficiently developed. Besides, the statement in the discussion (line 231-234) says it is upregulated in PTC's compared to normal, which is not the case in the supplementary Fig. 3?

Validity of the findings

The findings justify the relevance of this study in circRNA profiling of human PTC’s and its clinical implication as a maker.

---

## Round 0.3 · Minor Revisions

Dear Dr. Gang Yuan

Please proceed to the minor correction suggested by Reviewer 3 "correcting the spelling error that is on the name of the gene itself (most places-figure legend, figures, tables, results and discussion it was written as 'has_circRNA_' instead of 'hsa_circRNA_')."

Reviewer 1 ·

Basic reporting

The manuscript writing has been improved sufficiently.

Experimental design

The revised manuscript addressed all my previous concerns.

Validity of the findings

All concerns were addressed in this version.

Additional comments

The manuscript has been improved sufficiently from the previous version and the authors addressed all my concerns adequately

Reviewer 2 ·

Basic reporting

The authors addressed all the questions very carefully. English writing is improved in this revised version of the manuscript.

Experimental design

Everything is clearly mentioned.

Validity of the findings

The data is well organized and validated carefully. Quality of data is very clear and presentable.

Additional comments

The revised version the manuscript addressed all the questions. The quality of the manuscript is improved. I recommend this manuscript for publication.

Reviewer 3 ·

Basic reporting

Ren H. etal., by profiling circRNA expression identified differentially expressed circRNAs in PTC, of which the expression of hsa_circRNA_047771 was negatively correlated with BRAFV600E mutation, LNM and advanced TNM stage, whereas, the expression of hsa_circRNA_007148 was positively correlated with LNM.

The revised version of the manuscript has significantly improved and the authors have addressed most of the critics convincingly. Even though the functional/mechanistic aspects of the identified circRNA were not well established, I believe this manuscript would still benefit the recently evolving circRNA’s and their role in thyroid tumorigenesis. Hence, I recommend this manuscript may be published after correcting the spelling error that is on the name of the gene itself (most places-figure legend, figures, tables, results and discussion it was written as 'has_circRNA_' instead of 'hsa_circRNA_').

Experimental design

no comments

Validity of the findings

no comments.

---

## Round 0.4 · accepted · Accept

Dear Dr Gang Yuan,

Please in production, double check the "has_circRNA" in the title of Table 3 , for example" Association between hsa_circRNA_047771 and has_circRNA_007148 expression levels and
clinicopathological characteristics in40 PTC patients"

#